# Electrically-driven single-photon sources based on colloidal quantum dots with near-optimal antibunching at room temperature

Xing Lin[1], Xingliang Dai[2], Chaodan Pu[3], Yunzhou Deng[4], Yuan Niu[3], Limin Tong[1], Wei Fang[1], Yizheng Jin[4] & Xiaogang Peng[3]

Photonic quantum information requires high-purity, easily accessible, and scalable single-photon sources. Here, we report an electrically driven single-photon source based on colloidal quantum dots. Our solution-processed devices consist of isolated CdSe/CdS core/shell quantum dots sparsely buried in an insulating layer that is sandwiched between electron-transport and hole-transport layers. The devices generate single photons with near-optimal antibunching at room temperature, i.e., with a second-order temporal correlation function at zero delay ($g^{(2)}(0)$) being <0.05 for the best devices without any spectral filtering or background correction. The optimal $g^{(2)}(0)$ from single-dot electroluminescence breaks the lower $g^{(2)}(0)$ limit of the corresponding single-dot photoluminescence. Such highly suppressed multi-photon-emission probability is attributed to both novel device design and carrier injection/recombination dynamics. The device structure prevents background electroluminescence while offering efficient single-dot electroluminescence. A quantitative model is developed to illustrate the carrier injection/recombination dynamics of single-dot electroluminescence.

[1] Center for Chemistry of High-Performance & Novel Materials, State Key Laboratory of Modern Optical Instrumentation, College of Optical Science and Engineering, Zhejiang University, Hangzhou 310027, China. [2] Center for Chemistry of High-Performance & Novel Materials, State Key Laboratory of Silicon Materials, School of Materials Science and Engineering, Zhejiang University, Hangzhou 310027, China. [3] Center for Chemistry of High-Performance & Novel Materials, Department of Chemistry, Zhejiang University, Hangzhou 310027, China. [4] Center for Chemistry of High-Performance & Novel Materials, State Key Laboratory of Silicon Materials, Department of Chemistry, Zhejiang University, Hangzhou 310027, China. Xing Lin, Xingliang Dai and Chaodan Pu contributed equally to this work. Correspondence and requests for materials should be addressed to W.F. (email: wfang08@zju.edu.cn) or to Y.J. (email: yizhengjin@zju.edu.cn) or to X.P. (email: xpeng@zju.edu.cn)

Photons emitted from coherent or thermal emission sources follow a Poisson or super-Poisson distribution, which means that they inevitably arrive in bunches even when the average emission power is reduced to the single-photon level. In contrast, a single-photon source is a non-classical light source with sub-Poisson statistics, which ensures photons are emitted one by one (i.e., photons are "antibunched")[1, 2]. The auto-correlation function $g^{(2)}(0)$ is considered one of the most basic parameters for characterization of single-photon sources, with its value being <0.5 as a criterion for a quantum emitter in the single-photon regime. In fact, $g^{(2)}(0)$ vanishes for perfectly anti-bunched single photons[2].

Single-photon sources are indispensable for photonic quantum information technologies. Single-photon emission has been demonstrated in various systems[3–15]. For practical applications, compact as well as scalable single-photon sources, which can operate at room temperature and be electrically excited and controlled, are highly desirable[16]. In this regard, great efforts have been devoted to developing room-temperature and electrically driven single-photon sources using different material platforms, including defects in wide-bandgap inorganic semiconductors and organic molecules in organic host matrices[3–5]. However, $g^{(2)}(0)$ of any of these devices is far from being ideal.

Colloidal quantum dots are solution-grown semiconductor nanocrystals with sizes in the quantum-confinement regime[17]. Their size-dependent emission can cover a spectral range from ultra-violet to near infrared, including the commonly applied telecommunication wavelength window[18, 19]. Recently, near-unity photoluminescence quantum yield (QY), negligible dot-to-dot variation in photoluminescence peak width, and mono-exponential photoluminescence decay dynamics have been reproducibly achieved for well-developed quantum-dot systems,

such as CdSe/CdS core/shell ones[20–22]. The troublesome photo-luminescence blinking of a single-colloidal quantum-dot switch-ing between different brightness states under constant optical excitation–is greatly suppressed down to ~$10^{-5}$–$10^{-6}$ per photon absorption[20, 21, 23]. These advancements have enabled production of quantum-dot light-emitting diodes (LEDs) with near-unity internal quantum efficiency and long operational lifetime[24].

Here, we explore a single-colloidal quantum dot as a quantum emitter in electrically driven single-photon source. Combined with an isolating layer in the device, we realize near-optimal antibunching single-photon generation at room temperature with vanishing background emission. In addition, our quantitative model reveals that carrier injection/recombination dynamics suppress the multi-photon-emission probability.

## Results

**Device structure.** The structure of our electrically driven single-photon devices is illustrated in Fig. 1a. To fabricate such devices, poly(ethylenedioxythiophene):polystyrene sulfonate (PEDOT: PSS, ~30 nm), poly (N,N′-bis(4-butylphenyl)-N,N′-bis(phenyl)-benzidine) (Poly-TPD, ~30 nm), CdSe/CdS core/shell quantum dots (~0.1 dot μm$^{-2}$ of device area), poly (methyl methacrylate) (PMMA, ~12 nm), and ZnO nanoparticles (~50 nm) were deposited from solutions onto indium-tin-oxide (ITO)-coated glass substrates by spin-coating. Top silver electrodes were deposited by vacuum evaporation. Poly-TPD and ZnO were employed as hole-transport and electron-transport layers, respectively. The isolated quantum dots (~10 nm in diameters, Supplementary Fig. 1) were embedded in the insulating PMMA layer (Supplementary Fig. 2), which separated the ZnO electron-transport layer and the polymeric hole-transport layer.

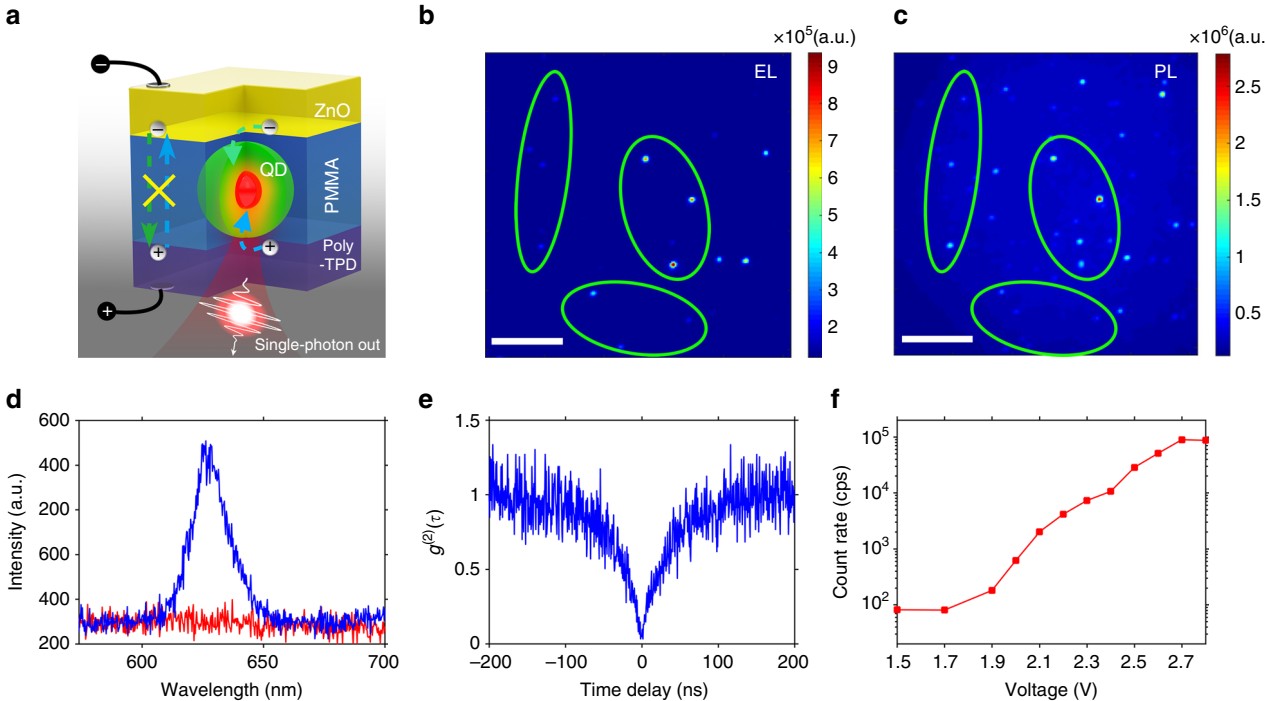

**Fig. 1** Optical properties of a single-photon device. **a** Schematic diagram of the key components of the device. **b**, Electroluminescence (driven at 2.4 V) and **c**, the corresponding photoluminescence (excited by 450 nm continuous wave laser) microscopic images. Scale bar: 5 μm. Green circles in **b** and **c** are shown as visual guides to highlight the relevant dots. **d** Electroluminescence spectrum (blue curve, driven at 2.8 V). The noise counts from the detector are also plotted (red curve). **e** $g^{(2)}(\tau)$ curve of a quantum-dot driven at 2.6 V. **f** Averaged electroluminescence counts of a single dot under different driving voltages (y-axis in log-scale)

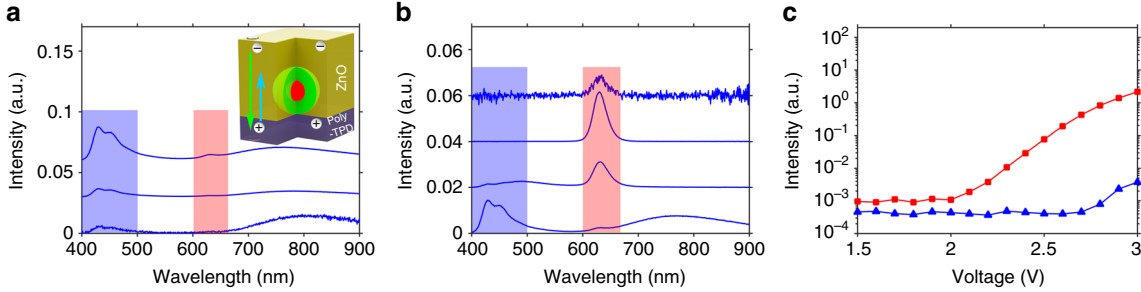

**Fig. 2** Optimization of the isolating film thickness. **a** Electroluminescence spectra of a device inheriting the conventional LED structure, i.e., isolated quantum dots directly sandwiched between the Poly-TPD and ZnO (without the PMMA layer, see inset for schematic drawing), at 3.0 V, 2.5 V, and 2.0 V (from top to bottom). The intensity of electroluminescence spectrum at 2.0 V is magnified by a factor of 20. **b** Electroluminescence spectra of the devices with isolated quantum dots embedded in PMMA layers with thicknesses of 17 nm, 12 nm, 10.4 nm and 0 nm, respectively (from top to bottom). The intensity of electroluminescence spectrum of device with a PMMA layer thickness of 17 nm is magnified by a factor of 50. The spectra are taken at a bias of 2.8 V. **c** Voltage-dependent integrated electroluminescence intensity from the isolated quantum dots (red squares) and Poly-TPD (blue triangles) in a device with a 12-nm PMMA layer. For quantitative comparison, all spectra were acquired from a 1-mm$^2$ emitting area of the devices. Red shaded areas (600–660 nm) and blue shaded areas (400–500 nm) in **a** and **b** illustrate the integration ranges used for quantum dot and Poly-TPD electroluminescence intensity calculations shown in (**c**), respectively

**Single-dot electroluminescence properties**. Electroluminescence of our devices was measured by a home-built fluorescence microscope system at room temperature. Figure 1b shows a false-color optical micrograph illustrating emission intensity of a device driven by a direct-current bias of 2.4 V. The bright spots in this image corresponded to electroluminescence from isolated quantum dots. With the photoluminescence micrograph image of the same area (Fig. 1c) as reference, electrical pumping at 2.4 V lit up about half of the CdSe/CdS core/shell quantum dots (Fig. 1b). The representative electroluminescence (Fig. 1d, under 2.8 V) and photoluminescence (Supplementary Fig. 3) spectra from the same bright spot exhibited similar peak position and peak contour, despite that the photoluminescence spectrum indicated background emission from Poly-TPD (Supplementary Fig. 3), while the electroluminescence spectrum was free of any background emission (Fig. 1d). Figure 1e shows the second-order temporal correlation function ($g^{(2)}(\tau)$) of a bright spot measured by the Hanbury Brown and Twiss method, which reveals a pronounced antibunching dip at zero delay. Quantitatively, $g^{(2)}(0)$ of this single-dot electroluminescence was determined to be $0.045 \pm 0.005$ without any spectral filtering or background correction (Fig. 1e). This value is much smaller than those of the reported electrically driven single-photon sources working at room temperature[3–5]. For electrically driven devices, such highly suppressed multi-photon-emission probability has only been achieved in the case of self-assembled quantum dots electro-excited at cryogenic temperature with temporal and spectral filtering[16, 25]. The average electroluminescence counts of our device (see Supplementary Fig. 4 for details) increased rapidly by increasing the voltage from 1.9 V to 2.8 V and reached ~89,000 counts s$^{-1}$ without background interference (Fig. 1f). Considering that the overall collection and detection efficiency of our optical system was ~8%, the total photon flux from the emitter was estimated to be 1,100,000 counts per second. The sub-bandgap turn-on voltage indicated efficient electrical driving of our devices. Multiple measurements on samples from different batches (Supplementary Fig. 5a) demonstrated that the antibunching photon emission from the isolated quantum dots was reproducible.

**Background-free single-photon emission with PMMA layer**. Our device design (Fig. 1a) played a key role in achieving such exceptionally low $g^{(2)}(0)$ shown in Fig. 1e. For all single-photon

sources, single-quantum emitters should be isolated from each other with typical distances larger than the corresponding emission wavelength. This feature is distinctly different from that of the LED devices. Consequently, inheriting the conventional quantum-dot LED structure for single-dot electroluminescent devices would cause direct contact of the electron-transport layer and hole-transport layer in most of the device area (>99.9%), which would lead to severe electron (hole) injection into the hole-transport layer (electron-transport layer) in device operation. This would cause troublesome background electroluminescence (Fig. 2a) and greatly degrade $g^{(2)}(0)$ of the single-dot electroluminescence. As shown in Fig. 2a, emission from Poly-TPD, which was observable at a bias voltage as low as 2.0 V, dominated the electroluminescence of the devices at all working voltages of 2.0–3.0 V. Furthermore, Supplementary Fig. 6 shows that almost all the isolated dots became invisible under electro-excitation when they were directly sandwiched between the hole-transport and electron-transport layers.

Embedding of isolated quantum dots in a PMMA layer between the hole-transport and electron-transport layers (Fig. 1a) allowed modulation of the electroluminescence of both single-dot and background emitters by controlling the thickness of the insulating layer (Fig. 2b). In principle, the PMMA layer should be sufficiently thick in order to block electron (hole) injection into the hole-transport layer (electron-transport layer) and prevent the isolated quantum dots from being quenched by direct contact with the ZnO nanoparticles[24]. However, if it was too thick, electroluminescence of a single-quantum dot would also be suppressed. A PMMA-thickness window of ~11–15 nm is identified to allow efficient single-dot electroluminescence without background emission under practical operating bias (Fig. 2b; Supplementary Figs. 7 and 8). For example, the device with a 12-nm PMMA layer exhibits background-free quantum-dot emission at 2.0–2.8 V (Fig. 2c).

The background-free single-photon emission achieved with our device structure (as shown in Figs. 1 and 2) is in stark contrast to the results obtained from other room-temperature operating electroluminescent devices. For instance, single-photon sources based on nitrogen-vacancy centers in diamonds exhibited a large $g^{(2)}(0)$ of 0.45, which was caused by significant background emission[3]. Regarding the device using organic quantum emitters embedded in a wide-bandgap polymer matrix (operating at 5 V), electroluminescence could take place either in

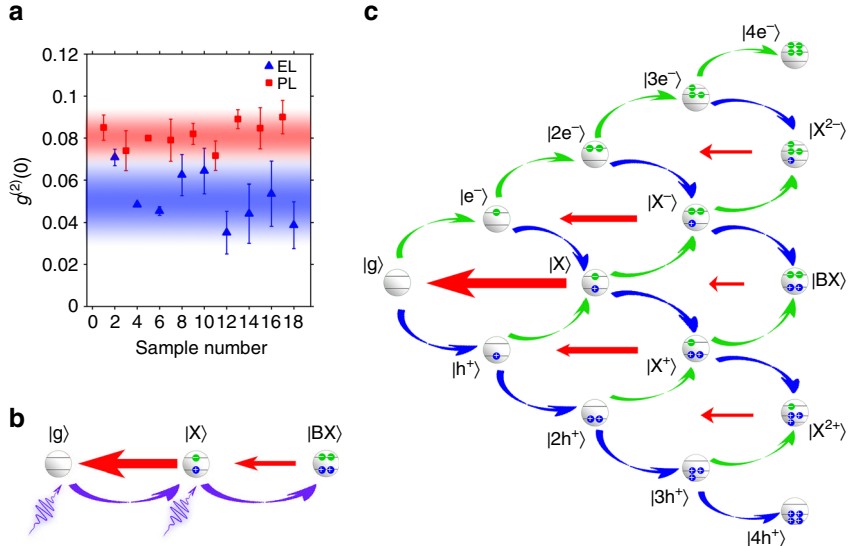

**Fig. 3** Single-dot electroluminescence and single-dot photoluminescence. **a** Lowest $g^2(0)$ values from the two sets of 9 quantum dots under electro-excitation (blue) and photo-excitation (red), respectively, The $g^{(2)}(0)$ value equals the average of four data points around zero time delay and error bars depict the standard error of these four measurements. **b** Carrier dynamics of photoluminescence for a non-blinking quantum dot. The violet curved arrows indicate the generation of a new electron–hole pair by absorbing a single excitation photon, while the straight red arrows indicate the recombination of one individual electron–hole pair. **c** Schematic drawings of carrier dynamics of electroluminescence. Here $|g\rangle$, $|e^-\rangle$, $|h^+\rangle$, $|ne^-\rangle$, $|nh^+\rangle$, $|X\rangle$, $|X^-\rangle$, $|X^+\rangle$, $|X^{2+}\rangle$, $|X^{2-}\rangle$, and $|BX\rangle$ represent ground state, single-electron state, single-hole state, n-electron state, n-hole state, exciton state, negatively charged-exciton state, positively charged-exciton state, three-hole-one-electron state, three-electron-one-hole state, and bi-exciton state, respectively. Note that the rare cases of one dot accumulating more than four carriers are ignored. The curved blue (green) up (down) arrows indicate the injection of a hole (electron) into a quantum dot

the polymer host or in the dopant organic quantum emitters[4]. As a result, background (host) emission was inevitable, which degraded the $g^{(2)}(0)$ of the organic single-photon emitter to a value > 0.9 without background correction. Literature also reported room-temperature photoluminescence of a single self-assembled quantum dot grown by molecular beam epitaxy, but the weak confinement of excitons within such dot reduces the photoluminescence QY at ambient temperature[26]. To our knowledge, the electrically driven single-photon source based on self-assembled quantum dots with demonstrated $g^2(0)$ of <0.1 could only work under stringent cryogenic temperatures with spectral filtering[16, 25].

**Single-dot electroluminescence vs. photoluminescence.** Interestingly, the $g^{(2)}(0)$ of our electrically driven devices can be statistically smaller than the lower limit of $g^{(2)}(0)$ of single-dot photoluminescence with isolated quantum dots deposited on background-free quartz substrates. As shown in Supplementary Fig. 5, all $g^{(2)}(0)$ values of the 26 photo-excited individual dots were higher than 0.07, while $g^{(2)}(0)$ values of 9 out of 30 electro-excited individual dots were below 0.07. The fact that $g^{(2)}(0)$ of single-dot electroluminescence could break the lower limit of $g^{(2)}(0)$ of background-free single-dot photoluminescence is illustrated by Fig. 3a, which plotted the smallest $g^{(2)}(0)$ values from the two sets of nine individual quantum dots under electro-excitation and photo-excitation, respectively.

To understand the origin of this interesting result, we compared carrier dynamics of single-dot photoluminescence and that of single-dot electroluminescence. For single-dot photoluminescence, the carrier dynamics can be described by a three-level system consisting of a ground state, as well as exciton and bi-exciton states (Fig. 3b). Generation of a bi-exciton from the ground state is a two-photon excitation process, i.e., simultaneous absorption of a second photon within the lifetime

of an exciton (~40 ns for our CdSe/CdS quantum dots, Supplementary Fig. 9). While the majority of bi-excitons decay non-radiatively via Auger recombination, a small amount of bi-excitons experience cascaded two-photon emission, which degrades antibunching of photons. Bawendi and co-workers demonstrated that the QY ratio between bi-excitons ($\eta_{BX}$) and excitons ($\eta_X$) imposed a lower limit for attainable $g^{(2)}(0)$ of single-dot photoluminescence under weak excitation[27], i.e., $g^{(2)}_{PL}(0) \geq \frac{\eta_{BX}}{\eta_X}$. Specifically, this lower limit was estimated to be 0.07 for the dots used in our experiment (Fig. 3a).

Carrier dynamics for single-dot electroluminescence is more complex than that of single-dot photoluminescence because the number of involved states is much greater and electro-excitation consists of two charge-injection channels. For the situation of up to four carriers, the carrier dynamics (Fig. 3c) is described by rate equations of the probability distribution concerning all 15 states (Supplementary Methods). An analogous argument for the analysis on single-dot photoluminescence gives an analytical expression of $g^{(2)}(0)$ under weak continuous electrical excitation as (refer to Supplementary Methods for details) $g^{(2)}_{EL}(0) = 0.41 \frac{k_X}{\bar{k}} \frac{\eta_{BX}}{\eta_X}$, where $k_X$ is the decay rate of the exciton state and $\bar{k}$ is defined by decay rates of the positive trion state ($k_Y$) and the negative trion state ($k_Z$) as $\bar{k} = \frac{k_Y k_Z}{k_Y + k_Z}$. Given that the decay rates of trion states are usually about one order of magnitude faster than that of exciton states due to the additional radiative recombination channel and non-radiative Auger-assisted recombination channel (Supplementary Fig. 9d and Supplementary Table 1), $\bar{k}$ is the significantly greater than $k_X$. Therefore, the lowest attainable $g^{(2)}_{EL}(0)$ is much smaller than that of $g^{(2)}_{PL}(0)$. The above analysis can be illustrated by the following physical picture. For both single-dot photoluminescence and single-dot electroluminescence, cascade emission of bi-excitons is responsible for the deterioration of antibunching of photons. An electro-generated bi-exciton can only be formed by injection of an

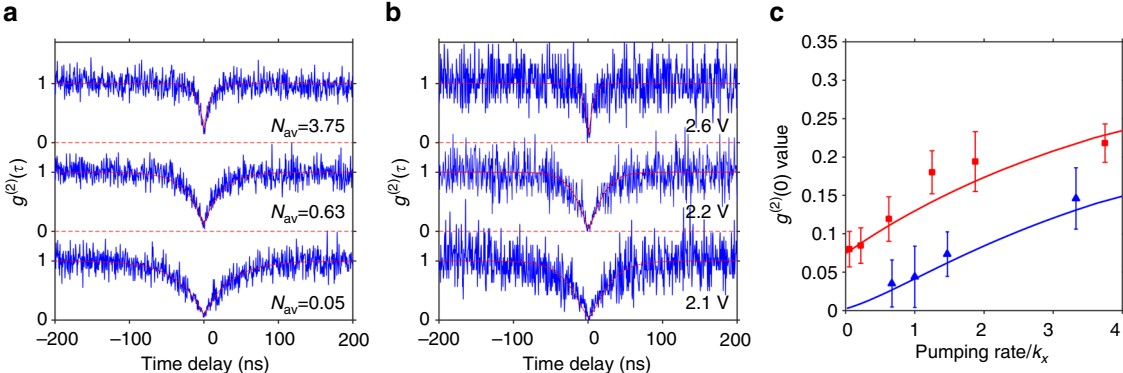

**Fig. 4** Single-dot $g^{(2)}(\tau)$ at different excitation conditions. **a,b** Antibunching curves of single-dot photoluminescence and electroluminescence at different excitation levels, respectively. $N_{av}$ in **a**, which is proportional to the excitation power, stands for the number of photons absorbed per lifetime cycle of excitons. **c** The dependence of simulated $g^{(2)}(0)$ on the relative pumping rates ($k_X = 25\,\mu s^{-1}$ is the decay rate of an exciton) for photoluminescence (red curve) and electroluminescence at the situation of balanced charge injection (blue curve). Red squares and blue triangles are experimental data for photoluminescence and electroluminescence from respective single-quantum dots, respectively. Standard deviations were calculated based on the values of four data points around zero time delay and plotted as error bars

electron into a positive trion or injection of a hole into a negative trion within the lifetime of the given trion state. Comparing to the lifetime of an exciton (~40 ns), the lifetime of the trion states (<1 ns for the $X^+$ and <5 ns for the $X^-$ for our dots, Supplementary Fig. 9d) is much shorter. In other words, the time window for the generation of a bi-exciton by electro-generation is about one-tenth of that by photo-generation. Thus the probability of the formation of electro-generated bi-excitons is significantly reduced, which improves the $g^{(2)}(0)$ of single-dot electroluminescence.

The above analysis was quantitatively modeled for both single-dot photoluminescence and single-dot electroluminescence. The parameters, including lifetimes and QYs of excitons, positive trions, negative trions, and bi-excitons (Supplementary Figs. 9–11) were determined by optical measurements. Both lifetime and QY of $X^-$ and $X^{++}$ were calculated according to a method developed by Sampat et al.[28]. We note that these lifetimes of the excited states from the optical measurements are used to represent the lifetimes in electroluminescence as an approximation. Regarding single-dot photoluminescence, the photon absorption rates, i.e., pumping rates, were estimated based on the knowledge of the quantum-dot absorption cross section[21] at given excitation wavelength and laser power. The $g^{(2)}(\tau)$ curves were produced by correlating the evolution of the three states in the three-level system to photon-detection events[27]. The simulated curves under different excitation powers matched well with the experimental ones (Fig. 4a). Regarding the single-dot electroluminescence, the principle to derive $g^{(2)}(\tau)$ curves is the same, i.e., describing carrier dynamics by rate equations of probability distribution of all the 15 involved states and then correlating the evolution of the states to photon-detection events. In this process, only the charge-injection rates are free fitting-parameters. We first considered the situation of balanced charge injection, i.e., the rate for electron and hole injection into a neutral dot being equal. For charged dots, Coulomb-interaction-modified injection rates were used (see Supplementary Methods and Supplementary Fig. 12). As shown in Fig. 4b, pumping rates (charge-injection rate into a neutral dot) of 16.8, 25.0, and 83.3 $\mu s^{-1}$ produced three $g^{(2)}(\tau)$ curves, which satisfactorily matched the antibunching curves of electroluminescence of a single dot at 2.1, 2.2, and 2.6 V biases, respectively. The modeling and experimental results further showed that, for both photoluminescence and electroluminescence, the lower the pumping rate was, the

lower the $g^{(2)}(0)$ value would become (Fig. 4c). The $g^{(2)}(0)$ for photoluminescence reached a lower limit of 0.07 under low-excitation conditions, while $g^{(2)}(0)$ for single-dot electroluminescence could be significantly lower than this limit. Further modeling results revealed that moderate deviation from the situation of balanced charge injection did not affect this important conclusion (Supplementary Fig. 13). The excellent match of the simulation results and our experimental data suggests that our model could well account the charge dynamics of electroluminescence of a single CdSe/CdS core/shell quantum dot.

## Discussion

Our work demonstrates a single-photon electroluminescence device which emits highly antibunched photons at room temperature. The excellent performance relies on the superior emissive properties of the CdSe/CdS core/shell quantum dots, the novel device structure which allows efficient electro-excitation of isolated quantum dots and excludes background emission from the hole-transport and electron-transport layers, and the suppression of bi-exciton formation due to the unique electro-excitation dynamics. In principle, this device structure and the solution-based processing are readily applicable to other colloidal quantum dots such as perovskite quantum dots[29] and quantum dots emitting at communication band[30]. Hereby, this work, accompanied with rapid synthetic development of colloidal quantum dots[31], highlights a pathway to the development of novel room-temperature quantum light sources for practical quantum information applications.

## Methods

**Synthesis of CdSe–CdS core/shell quantum dots.** CdSe-core dots with first absorption peak at 550 nm were synthesized by using a literature method[32] with some modifications. In a typical synthesis, 0.2 mmol of cadmium stearate (CdSt$_2$) and 3.5 mL of 1-octadecene (ODE) were loaded into a 25 mL three-neck flask. After argon bubbling, Se-ODE suspension (0.05 mmol of elemental selenium dispersed in 0.5 mL of ODE) was injected into the reaction flask at 250 °C. After growth for ~8 min, Se-ODE suspension was added dropwise until the first absorption peak of the CdSe dots reached 550 nm. The reaction mixture was allowed to cool down to 50 °C. An in situ extraction procedure was carried out to remove unreacted precursors and side products[33]. For the growth of the CdS shell, 1.2 mL of dodecane, 3.8 mL of oleylamine, and 1 mL of purified CdSe-core solution (~3 × 10$^{-7}$ mol of CdSe dots) were loaded. This mixture was heated to 80 °C for the addition of cadmium diethyldithiocarbamate (Cd(DDTC)$_2$) and then heated to 160 °C for the growth of CdS. The time for the growth of first monolayer of CdS was 40 min. The time for the growth of all other monolayers of CdS was 20 min. This temperature

cycle was applied for the growth of every monolayer of CdS. The amount of the precursor solution for ten consecutive injections was calibrated as 0.08, 0.11, 0.15, 0.20, 0.26, 0.32, 0.39, 0.46, 0.54, and 0.63 mL respectively. For the first six monolayers of CdS, $Cd(DDTC)_2$ was used as the sole precursor. From the seventh monolayer of the CdS shell, the precursor solution was changed to a mixture of Cd$(DDTC)_2$ and cadmium oleate with a molar ratio of 4:1. The CdSe/CdS core/shell quantum dots were precipitated by ethanol and dispersed in a mixture of octane and 2-ethyl-1-hexanethiol with a volume ratio of 1:1. The ligand exchange process lasted for 1 h, and the resulting CdSe/CdS core/shell quantum dots were precipitated by ethanol and re-dispersed in octane.

**Device fabrication**. The devices were fabricated by depositing materials onto ITO-coated glass substrates (0.18 mm in thickness and ~100 Ω per square in sheet resistance). PEDOT: PSS solutions (BaytronP VP Al 4083, filtered through a 0.45 mm N66 filter) were spin-coated onto the substrates at 4000 r.p.m. for 50 s and baked at 150 °C for 15 min. Next the PEDOT:PSS coated substrates were transferred to a $N_2$-filled glove box ($O_2$ < 1 ppm, $H_2O$ < 1 ppm). Solutions of poly(N,N′-bis(4-butylphenyl)-N,N′-bis(phenyl)-benzidine) (Poly-TPD in chlorobenzene, 8 mg mL$^{-1}$), quantum dots (in octane), PMMA (in acetone, 1.0 mg mL$^{-1}$), and ZnO nanoparticles (in ethanol, ~40 mg mL$^{-1}$) were used for the fabrication of the devices. The quantum-dot solution (optical density: 1.0 at 400 nm) was diluted 50,000-fold before processing. For the results shown in Fig. 2, and Supplementary Fig. 6, the devices were fabricated using quantum-dot solutions with a concentration of 10 times higher. AFM measurements showed that the quantum dots are still isolated from each other in these devices. All layers other than PEDOT: PSS were spin-coated at 2000 r.p.m. for 45 s. The poly-TPD layer was baked at 130 °C for 30 min before the deposition of the next layer. Finally, the multi-layered samples were loaded into a thermal evaporator (Trovato 300C) with a base pressure of ~2 × 10$^{-7}$ Torr for deposition of top silver electrodes (100 nm). The devices were encapsulated in the glove-box by the cover glasses using ultraviolet-curable resin.

**Optical measurements**. All measurements were carried out at room temperature (20–22 °C) and ambient conditions. Electroluminescence and photoluminescence from colloidal quantum dots were characterized by a home-built fluorescence microscope system. Samples were mounted on a piezoelectric XYZ stage designed for precise position controlling. A direct-current source (Keithley 2400) was used for electroluminescence and a laser diode (PicoQuant LHD-450) was used for photoluminescence. Emissions from quantum dots were collected by an oil immersion objective with a N.A. of 1.46. The luminescence was either directed to an electron multiplier charge coupled device (EMCCD, Andor iXon3) for imaging or detected by avalanche photodetectors (SPCMs, PerkinElmer SPCM-AQRH-15-FC) through a 9-µm (core diameter) single-mode fiber (Thorlabs SM28e) pinhole. Photon correlation measurements were performed using a time-correlated single-photon counting module (PicoHarp 300) with a 50:50 beam splitter. Emission spectra were recorded by a spectrometer (Andor Shamrock 303i). For the measurements to produce Fig. 2 and Supplementary Fig. 7, the spectra were acquired by a fiber-based spectrometer (Ocean Optics, QEPro) to get large-area (~1 mm$^2$) and averaged results. In the photoluminescence experiments, a long-pass filter (561 nm) was used to block the light from the pumping laser.

**Data availability**. The data that support the findings of this study are available from the corresponding authors on request.

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

## Acknowledgements

We thank Dr. Arash Rahimi-Iman for manuscript proofreading and helpful suggestions. This work was financially supported by the National Key R&D Program of China (2016YFB0401600), the National Basic Research Program of China (2014CB921303), the National Natural Science Foundation of China (51522209, 91433204, 21233005, 61635009), and the Fundamental Research Funds for the Central Universities (2015FZA3005).

## Author contributions

W.F., Y.J., and X.P. conceived the design of the devices. X.L. built the setup for optical characterizations and carried out the single-photon measurements and modeling of charge dynamics under the supervision of W.F. and X.P. X.D. carried out the device fabrication under the supervision of Y.J. C.P. and Y.N. synthesized and modified the quantum dots under X.P.'s supervision. Y.D. carried out the atomic force microscopy measurements. L.T. offered useful discussions for carrying out the work. W.F., Y.J., and

X.P. wrote the manuscript. All authors discussed the results and commented on the manuscript.

## Additional information

**Competing interests:** The authors declare no competing financial interests.

