## [Peer Review File · Nature Communications]

Reviewers' comments:

Reviewer #1 (Remarks to the Author):

The manuscript demonstrated room temperature single photon emission from electrically driven single colloidal quantum dots. Both high temperature and electrical operation is important for realizing practical quantum light sources. Although electrically driven single photon emission at room temperature have been demonstrated from other platforms such as epitaxial grown quantum dots and defects in diamond, it have not been demonstrated from the colloidal quantum dots, yet. This manuscript is worth to be published in a high impact journal, but need some improvement before it can be accepted to nature communications:

(1)The authors try to overinterpret their results to explain their observation. The explanation of the improvement of $g_2(0)$ due to different carrier recombination dynamics between electroluminescence and photoluminescence is not clear. Also I cannot see any significant improvements of single photon properties in electroluminescence compared to photoluminescence, but the authors compare these two excitation conditions in several parts of the manuscript. Only the reduction of background emission by electroluminescence looks an important difference. Why do the authors focus on these comparisons, which have very small differences, and the origin is not clear. Why is the comparison important?

I list detailed comments on the manuscript as follows.

(2) The authors claim that the origin of background is from poly-TPD, but a ZnO layer also have defect emission at the similar wavelength around 600 nm (ACS Appl. Mater. Interfaces, 7, 5619 (2015)). How do the authors determine the origin of the background emission?

(3) The difference of $g_2(0)$ between electroluminescence (0.08 ± 0.04) and photoluminescence (0.11 ± 0.02) does not seem to be remarkable as the authors claim on page 8. Their difference is within an error range, and why is this small difference important?

(4) The authors claim that the different carrier dynamics between electro-and photo-excitation is the origin of the difference in $g_2(0)$, but as the authors show in figure S9, in general, carrier dynamics with photo-excitation is not as simple as figure 3b. The explanation correlating the carrier dynamics and $g_2(0)$ raise a question. If simple carrier dynamics in non-blinking quantum dots causes poor $g_2(0)$, do blinking quantum dots with complicated carrier dynamics have better $g_2(0)$? This is a strange tradeoff between non-blinking and $g_2(0)$. Also, the explanation is not sufficient to understand how more complicated carrier dynamics improves $g_2(0)$. The formation of multiple excitons such as trions with a fast lifetime (~ 1 ns) can also degrade $g_2(0)$ due to their multiple excitation and emission within a long lifetime of excitons (40 ns).

(5) The authors fit the $g_2(t)$ data with multiple measured and calculated parameters as well as several free (?) fitting parameters explained on page S5. What is the parameter set, $\sigma_1, \sigma_2, \sigma_3, \sigma_4, \sigma_5, \sigma_6$, and another parameter set, $\sigma_5, \sigma_8, \sigma_9, \sigma_{13}, \sigma_{12}, \sigma_{14}$. Are they free fitting parameters for electroluminescence? In general, the second-order correlation function of single quantum dots, $g_2(t)$ can be fitted with a single parameter (lifetime). If one try to fit $g_2(0)$ curves with multiple fitting parameters, it would always fit the data well, but I don't agree that this is meaningful. Furthermore, how can the measured lifetimes of X, X-, X+, and BX from photoluminescence represent well the actual lifetimes in electroluminescence, in particular when electroluminescence and photoluminescence have different carrier dynamics. Also lifetime varies emitter to emitter.

Reviewer #2 (Remarks to the Author):

Authors have answered my previous questions. I am happy to recommend its publication after the following revision.

The title is misleading. Electrically driven Single photon source with near perfect g_2 has been demonstrated already (see Nature Photonics 6, 299–303 (2012)]. "Colloidal quantum dots" must go to the title to avoid misleading.

Reviewer #3 (Remarks to the Author):

The revised manuscript and supporting material appears to have been thoroughly reworked by the authors. More technical information is now provided to support some of the claims better. The overall quality has also been improved. Although I am somewhat skeptical of some of the statements made, I believe the manuscript can be published in Nature communications.

Point-by-Point Response to the Reviewers

Reviewer #1

Comment 1: *The manuscript demonstrated room temperature single photon emission from electrically driven single colloidal quantum dots. Both high temperature and electrical operation is important for realizing practical quantum light sources. Although electrically driven single photon emission at room temperature have been demonstrated from other platforms such as epitaxial grown quantum dots and defects in diamond, it have not been demonstrated from the colloidal quantum dots, yet. This manuscript is worth to be published in a high impact journal, but need some improvement before it can be accepted to nature communications: (1) The authors try to over interpret their results to explain their observation. The explanation of the improvement of $g^{(2)}(0)$ due to different carrier recombination dynamics between electroluminescence and photoluminescence is not clear. Also I cannot see any significant improvements of single photon properties in electroluminescence compared to photoluminescence, but the authors compare these two excitation conditions in several parts of the manuscript. Only the reduction of background emission by electroluminescence looks an important difference. Why do the authors focus on these comparisons, which have very small differences, and the origin is not clear. Why is the comparison important?*

Our revision and responses: We appreciate the reviewer for the overall positive feedback on our work. The reviewer's concerns on the comparisons of the $g^{(2)}(0)$ and charge dynamics of single-dot electroluminescence and single-dot photoluminescence motivated us to re-examine the experimental results and the corresponding physical models. These efforts lead to a concise analytical expression for the lower limit of $g^{(2)}(0)$ of single-dot electroluminescence. Comparing this analytical expression with that for single-dot photoluminescence indicates that the attainable value of $g^{(2)}(0)$ of single-dot electroluminescence is much smaller than that of single-dot photoluminescence, which is supported by experimental results. We provide detailed justifications below.

For single-dot photoluminescence, Bawendi and co-workers developed a model (Ref. 28), which describes carrier dynamics by rate equations of probability distribution of all states in a three-level system, as schematically shown in Figure 3b. The evolution of the probability distribution of the

states leads to an analytical solution of $g^{(2)}(\tau)$. Under weak continuous excitation (i.e. $P/k_x \rightarrow 0$, where P is pumping rate and k_x is exciton decay rate), $g^{(2)}(0)$ approaches a lower limit:

$$g_{PL}^{(2)}(0) = \frac{\eta_{BX}}{\eta_X}$$

where η_{BX} and η_X are the quantum yields of biexciton and exciton, respectively.

The principle of our approach to derive $g^{(2)}(\tau)$ for single-dot electroluminescence is the same, i.e. describing carrier dynamics by rate equations of probability distribution of all the involved states (15 states in total) and then correlating the evolution of the states to photon-detection events to acquire $g^{(2)}(\tau)$ curves. Our physical model for single-dot electroluminescence is more complex than the model for single-dot photoluminescence because the number of involved states is greater than that of the 3-level system and electro-excitation of quantum dots consists of two charge injection channels. However, an analogous argument to the analyses on single-dot photoluminescence gives an analytical expression of $g^{(2)}(0)$ under weak continuous electrical excitation (i.e. $P/k_x \rightarrow 0$) as (please refer to revised **Supplementary Information (S3)** for details) :

$$g_{EL}^{(2)}(0) = \frac{\beta k_X \eta_{BX}}{4 \bar{k} \eta_X}$$

where β is a constant defined by the modification coefficient γ as:

$$\beta = \gamma(1 + 2\gamma + 2\gamma^4 + 2\gamma^9 + 2\gamma^{16})$$

which equals to ~ 1.643 as we assign 0.63 to γ (See SI Section 2),

and \bar{k} is defined by decay rates of X^+ and X^- (k_Y and k_Z) as:

$$\bar{k} = \frac{k_Y k_Z}{k_Y + k_Z}$$

We emphasize that this lower limit of $g_{EL}^{(2)}(0)$ under weak continuous electrical excitation ($P/k_x \rightarrow 0$) is an analytical solution derived without any free or fitting parameters. The low limit of $g_{EL}^{(2)}(0)$ equals to that of $g_{PL}^{(2)}(0)$ multiplied by a factor of $0.4k_X/\bar{k}$. Thus the lowest attainable $g_{EL}^{(2)}(0)$ is much smaller than that of $g_{PL}^{(2)}(0)$ since the decay rates of trion states are usually about one order of magnitude faster than that of exciton.

The above analyses are supported by our experimental results. The difference between the measured $g^{(2)}(0)$ from single-dot electroluminescence and those from single-dot photoluminescence

is evident as shown by the $g^{(2)}(\tau)$ curves in Fig. S5, regardless of the calculation method for $g^{(2)}(0)$. We plotted the lowest $g^{(2)}(0)$ values from two sets of 9 quantum dots under electro-excitation (blue) and photo-excitation (red), respectively. The distribution can be divided into two groups, with $g^{(2)}(0)$ values from photo-excited ones on the top (≥ 0.07) and those from electro-excited ones on the bottom (≤ 0.07) (as shown in Fig. 3a in the revised manuscript, also shown below).

Figure 3. Single-dot electroluminescence and single-dot photoluminescence. **a**, Lowest $g^{(2)}(0)$ values from the two sets of 9 quantum dots under electro-excitation (blue) and photo-excitation (red), respectively. The $g^{(2)}(0)$ value was obtained from the average of the 4 data points around zero time delay of the measured $g^{(2)}(\tau)$ curve and error bar comes from the standard error of these 4 data points. **(b and c**, same as previous version).

We believe that the interesting finding of the $g^{(2)}(0)$ of single-dot electroluminescence can be smaller than that of single-dot photoluminescence is worth detailed discussions. In many cases, the $g^{(2)}(0)$ of electroluminescence from isolated single quantum emitters are poorer than those of the corresponding photoluminescence because of background emission (Ref. 2). In our work, the novel device structure prevents background electroluminescence, as the reviewer has already recognized. Note that we compared the $g^{(2)}(0)$ values of single-dot electroluminescence and those of single-dot photoluminescence with isolated single dots on quartz substrates, which are virtually background free. Therefore, the cause of no background emission is not sufficient to explain this interesting fact and the mechanism based on different charge dynamics must be invoked.

Comment 2: I list detailed comments on the manuscript as follows.
(2) The authors claim that the origin of background is from poly-TPD, but a ZnO layer also have defect emission at the similar wavelength around 600 nm (ACS Appl. Mater. Interfaces, 7, 5619 (2015)). How do the authors determine the origin of the background emission?

Our revision and responses: We measured photoluminescence spectra of both Poly-TPD and ZnO under conditions similar to the single-dot photoluminescence experiments, i.e. a 450 nm CW laser and 40 Wcm^{-2} , as shown below. Within the spectral range under consideration (600-650 nm), ZnO (produced in our labs) shows negligible emission comparing to that from Poly-TPD. Therefore we conclude that the emission from Poly-TPD dominates the background emission. We added this plot as Fig. S3c in the revised Supplementary Information.

Figure S3c. Photoluminescence spectra of Poly-TPD (black curve) and ZnO (red curve) under same optical excitation conditions as b. Within the spectral range under consideration (600-650 nm), ZnO shows negligible emission comparing to that from Poly-TPD.

Comment 3: (3) The difference of $g_2(0)$ between electroluminescence (0.08 ± 0.04) and photoluminescence (0.11 ± 0.02) does not seem to be remarkable as the authors claim on page 8. Their difference is within an error range, and why is this small difference important?

Our revision and responses: As we have discussed in the responses to **comment 1**, two sets of photo-excited and electro-excited quantum dots with lowest $g^{(2)}(0)$ values show distinct differences on the $g^{(2)}(0)$ distributions. So we replaced Fig. 3a in revised manuscript to highlight such differences.

Regarding the statistical results for the $g^{(2)}(0)$ of all electro-excited single dots (0.08 ± 0.04), the standard deviation is relatively large. One of the major reasons for such spread distribution of $g^{(2)}(0)$ is due to fluctuation of the thicknesses of PMMA layer, which modulates electron injection rates and thus affects charge balance. At optimal conditions, the EL intensity from single quantum dot shows non-blinking-like time trace at low applied voltages, as shown in Fig. S4. In such cases, the $g^{(2)}(0)$ values are generally smaller than the lower limit of $g^{(2)}(0)$ value (~ 0.07) from photo-excited quantum dots. We would like to note that the uncertainty of the $g^{(2)}(0)$ values has been taken into consideration in such comparison (e.g., the $g^{(2)}(0)$ value shown in Fig. 1e is 0.045 ± 0.005 without any background subtraction). At un-optimal conditions, the unbalanced charge injection results in long-lived charged states in the quantum dots which reduce the average intensity. As a consequence, the signal to noise (mainly come from dark counts of APD) ratio is diminished. Considering that the coincident counts at $\tau = 0$ is almost zero, the diminished signal to noise ratio degrades the $g^{(2)}(0)$ values, which brings wide distributions of the statistical results.

Comment 4: (4) *The authors claim that the different carrier dynamics between electro-and photo-excitation is the origin of the difference in $g_2(0)$, but as the authors show in figure S9, in general, carrier dynamics with photo-excitation is not as simple as figure 3b. The explanation correlating the carrier dynamics and $g_2(0)$ raise a question. If simple carrier dynamics in non-blinking quantum dots causes poor $g_2(0)$, do blinking quantum dots with complicated carrier dynamics have better $g_2(0)$? This is a strange tradeoff between non-blinking and $g_2(0)$.*

Our revision and responses: We are sorry for omitting the conditions for the two sets of photo-excitation experiments, which may cause misunderstanding. The excitation power ($N_{av} = 0.6$) for the measurements shown in Fig. S9a-d is much higher than the ones we used for $g^{(2)}(0)$ measurements. For example, the typical excitation power used to measure the $g^{(2)}(0)$ values (for the experiments shown in Fig. S6b) has a N_{av} of < 0.1 . As shown in Fig. S9e, under excitation conditions of $N_{av} < 0.3$, the quantum dots show virtually non-blinking time trace. For the experiments to produce Fig. S9a-d,

we applied a high excitation power to generate sufficient trion states so that the parameters such as decay rates of X^- and X^+ can be extracted.

Based on our experimental results, the blinking quantum dots under high optical excitation power usually exhibit larger $g^{(2)}(0)$ comparing to those from quantum dots under low excitation power which are non-blinking. The $g^{(2)}(0)$ values reach minimum at lower excitation power ($N_{av} \rightarrow 0$) in our experiments (red dots in Fig. 4c), which coincides with theoretical predictions based on the model shown in Fig. 3b.

Comment 5: *Also, the explanation is not sufficient to understand how more complicated carrier dynamics improves $g^{(2)}(0)$. The formation of multiple excitons such as trions with a fast lifetime (~ 1 ns) can also degrade $g^{(2)}(0)$ due to their multiple excitation and emission within a long lifetime of excitons (40 ns).*

Our revision and responses: We thank the reviewer for this critical comment that not only the formation of bi-exciton, but also the multiple excitation and emission of quantum dots may degrade $g^{(2)}(0)$. We would like to clarify that in theory, the emission from a charged exciton state (i.e., radiative decay process such as $|X^+\rangle \rightarrow |h^+\rangle$, $|X^{2+}\rangle \rightarrow |2h^+\rangle$, etc.) would NOT degrade $g^{(2)}(0)$ due to the fact that the follow-up state ($|h^+\rangle$, $|2h^+\rangle$, etc.) does not emit another photon immediately. In this sense, the only state that spoils the $g^{(2)}(0)$ value is the bi-exciton state. Experimentally, when a finite time resolution of our measurement system is taking into consideration, it is possible that the multiple excitation and emission from trion states degrade the measured $g^{(2)}(0)$ value, due to a finite possibility of multi-photon emission within system resolution time. However, the key parameter is not the lifetimes of trion states. For example, considering a process that starts from the $|X^+\rangle$ state, the multi-excitation and multi-emission process may evolve as $|X^+\rangle \xrightarrow{h\nu} |h^+\rangle \xrightarrow{+e} |X\rangle \xrightarrow{h\nu} |ground\rangle$. For HBT measurement, $\tau = 0$ when the first photon emitted from $|X^+\rangle$ is detected by one of the detector. The average time for the second photon to arrive at the second detector is determined by the sum of electron injection rate and exciton decay rate (on the time scale of $1/(k_x + P_e)$, where P_e is electron injection rate). Under weak excitation conditions the average arrival time of the second photon converges to exciton lifetime (40 ns), which is much longer comparing to the system resolution time (~ 500 ps). In fact, this interval is exactly the same to that of photo-excitation if the

multi-excitation and multi-emission process starts from an exciton state. Therefore in both single-dot electroluminescence and single-dot photoluminescence, the process of multiple excitation and emission do not significantly degrade $g^{(2)}(0)$.

We would like to point out that the short lifetime of trion state is the key factor which makes the lower limit of $g^{(2)}(0)$ of single-dot electroluminescence much smaller than that of single-dot photoluminescence. As we have discussed in the responses to **comment 1**, as well as in the revised **Supplementary Information** (S3), the analytical expression for the lower limit of $g^{(2)}(0)$ of electro-excitation differs from that of photo-excitation by a factor of $\frac{\beta k_X}{4 \bar{k}}$, where $\bar{k} = \frac{k_Y k_Z}{k_Y + k_Z}$ implies the decay rates of two trion states. This means that faster trion decay rates (or shorter trion lifetimes) shall result in smaller $g^{(2)}(0)$ of single-dot electroluminescence. This fact can be understood by the following physical picture schematically shown in Fig. 3b-c. The exciton and trion states (X^+ and X) are the corresponding preceding states to form BX for photo-excitation and electro-excitation, respectively. Under equivalent excitation power (i.e., similar rate of absorbing a single photon or injection a single electron/hole), the probability to form BX by electro-excitation is much smaller than that by photo-excitation, because the lifetime of the trion states (~ 1 ns) is more than one order of magnitude shorter than that of the exciton state (~ 40 ns). Given the fact that bi-exciton state is the main source that spoils the $g^{(2)}(0)$ value, the lower limit of $g^{(2)}(0)$ of single-dot electroluminescence is much smaller than that of single-dot photoluminescence.

Comment 6: (5) *The authors fit the $g_2(t)$ data with multiple measured and calculated parameters as well as several free (?) fitting parameters explained on page S5. What is the parameter set, $\sigma_1, \sigma_2, \sigma_3, \sigma_4, \sigma_5, \sigma_6$, and another parameter set, $\sigma_5, \sigma_8, \sigma_9, \sigma_{13}, \sigma_{12}, \sigma_{14}$. Are they free fitting parameters for electroluminescence? In general, the second-order correlation function of single quantum dots, $g_2(t)$ can be fitted with a single parameter (lifetime). If one try to fit $g_2(0)$ curves with multiple fitting parameters, it would always fit the data well, but I don't agree that this is meaningful. Furthermore, how can the measured lifetimes of X, X^-, X^+ , and BX from photoluminescence represent well the actual lifetimes in electroluminescence, in particular when electroluminescence and photoluminescence have different carrier dynamics. Also lifetime varies emitter to emitter.*

Our revision and responses: $\sigma_0 \sim \sigma_{15}$ are the time-dependent variables in rate equations standing for the population of 15 states. With proper initial conditions ($\sigma(t=0)$) and carrier injection rate provided, we can calculate how carriers in a single quantum dot evolve with time, as well as photon emission events. In this way, we can also calculate and plot $g^{(2)}(\tau)$ curves under different experimental conditions. For experimentally measured $g^{(2)}(\tau)$ curves shown in Fig. 4b, the carrier injection rate, which is determined by the applied voltage, is the only fitting parameter (assuming balanced electron/hole injection) for the curve fitting, as we cannot directly measure this parameter in a device. There is no other free parameter in the rate equations. We have obtained a number of coefficients by optical measurements or calculated the rest of the coefficients we used in the rate equations (as listed in Table S1). We agree with the reviewer that the actually lifetimes in electroluminescence are not exactly same as those in photoluminescence. And CW electro-excitation operation condition makes the lifetime measurements of X^+ and X^- states difficult. Here we take the lifetimes measured from photoluminescence as an approximation. We have added in the revised manuscript “We note that these lifetimes of the excited states from the optical measurements are used to represent the lifetimes in electroluminescence as an approximation.” in the second paragraph on page 10.

We agree with the reviewer that the single parameter (single exciton decay time) fitting for $g^{(2)}(\tau)$ is widely used in PL measurements in the form of $1-a*\exp(-\tau/\tau_x)$ (Ref. 10,11). This is because for non-blinking quantum dots, majority of the photons collected are emitted from exciton states. However, if the emission process involves several other states, single parameter fitting based on exciton lifetime (τ_x) is no longer valid. Particularly, in the cases of electro-excitation in systems such as self-assembled QDs (Ref. 13), SiC (Ref. 4) defects and NV center in diamond (Ref. 2), equations with different forms have to be applied due to the complicate carrier dynamics and other intermediate states. In our case, the existence of two carrier injection channels makes the emission from X^+ and X^- states significant, especially when a high driving voltage is applied. This render the prerequisites of a fitting simply based on exciton lifetime and the carrier dynamics is too complicate to be expressed by a simple formula. We have to calculate numerically.

We believe that the approximation of using the coefficients determined by optical measurements shall not violate the fact that the lifetimes of trion states (X^+ and X^-) are much shorter than that of exciton under electro-excitation. As we have discussed above, this holds the key to realize near-optimal antibunching of single-photon emission in our device. Therefore, we believe that the general conclusion of the lower limit of $g^{(2)}(0)$ of single-dot electroluminescence is smaller than that of single-dot photoluminescence is still valid.

Reviewer #2

Comment 1: *Authors have answered my previous questions. I am happy to recommend its publication after the following revision. The title is misleading. Electrically driven Single photon source with near perfect g_2 has been demonstrated already (see Nature Photonics 6, 299–303 (2012)). "Colloidal quantum dots" must go to the title to avoid misleading.*

Our revision and responses: We thank the reviewer for the recommendation for publication. The title has been changed as “Near-optimal-Antibunching, Electrically-driven, and Room-temperature Single-photon Sources Based on Colloidal Quantum Dots”.

Reviewer #3

Comment 1: *The revised manuscript and supporting material appears to have been thoroughly reworked by the authors. More technical information is now provided to support some of the claims better. The overall quality has also been improved. Although I am somewhat skeptical of some of the statements made, I believe the manuscript can be published in Nature communications.*

Our revision and responses: We thank the reviewer for the nice comment.

List of major changes:

1. According to reviewer #2's suggestion, we change the title as "Near-optimal-Antibunching, Electrically-driven, and Room-temperature Single-photon Sources Based on Colloidal Quantum Dots"
2. To demonstrate the distinct differences between the $g^{(2)}(0)$ of single-dot electroluminescence and that of single-dot photoluminescence, we replace Fig. 3a with a new plot and move the original one as Fig. S5c. The paragraph describing Fig. 3a has been modified accordingly, as shown on pp.7-8.
3. Analytical expression of the $g^{(2)}(0)$ of single-dot electroluminescence under weak excitation condition is developed (**Supplementary Information (S3)**) and the paragraph to discuss carrier dynamics of electroluminescence is completely rewritten (pp. 8-10). For comparison, the analytical expression of $g^{(2)}(0)$ of single-dot photoluminescence is also presented in the revised manuscript (second paragraph on page 8).
4. Part of the second paragraph on page 10 is rephrased to make the explanation of Fig. 4 clearer. And the annotations of calculated $g^{(2)}(0)$ curves in Fig. 4c are corrected as "PL modeling" and "EL modeling" for photoluminescence and electroluminescence, respectively, as the curves are calculated directly from the parameters determined from optical measurements rather curve fitting from the experimental data shown in Fig. 4c.
5. Photoluminescence spectra of Poly-TPD and ZnO under same optical excitation conditions are presented in Fig. S3c, which indicates the major contribution of photoluminescence background emission is from Poly-TPD.

REVIEWERS' COMMENTS:

Reviewer #1 (Remarks to the Author):

The authors have addressed all of my concerns. The paper can be published.